# Trophic niche overlap between coyotes and gray foxes in a temperate forest in Durango, Mexico

César Ricardo Rodríguez-Luna[1]☯, Jorge Servín [2]☯, David Valenzuela-Galván [3]☯*, Rurik List [4]☯

1 Doctorado en Ciencias Biológicas y de la Salud, Universidad Autónoma Metropolitana, Ciudad de México, Mexico, 2 Departamento El Hombre y su Ambiente, División de Ciencias Biológicas y de la Salud, Universidad Autónoma Metropolita–Unidad Xochimilco, Coyoacán, Mexico City, Mexico, 3 Departamento de Ecología Evolutiva, Centro de Investigación en Biodiversidad y Conservación, Universidad Autónoma del Estado de Morelos, Cuernavaca, Morelos, Mexico, 4 Departamento de Ciencias Ambientales, División de Ciencias Biológicas y de la Salud, Universidad Autónoma Metropolitana–Unidad Lerma, Lerma de Villada, Mexico, Mexico

☯ These authors contributed equally to this work.
* dvalen@uaem.mx

**Data Availability Statement:** All relevant data are within the manuscript and its Supporting Information files.

## Abstract

Resource partitioning, and especially dietary partitioning, is a mechanism that has been studied for several canid species as a means to understand competitive relationships and the ability of these species to coexist. Coyotes (*Canis latrans*) and gray foxes (*Urocyon cinereoargenteus*) are two canid species that are widely distributed, in Mexico, and they are sympatric throughout most of their distribution range. However, trophic dynamic and overlap between them have not been thoroughly studied. In order to better understand their ecological relationship and potential competitive interactions, we studied the trophic niche overlap between both canids in a temperate forest of Durango, Mexico. The results are based on the analysis of 540 coyote and 307 gray fox feces collected in 2018. Both species consumed a similar range of food items, but the coyote consumed large species while the gray fox did not. For both species, the most frequently consumed food categories throughout the year and seasonally were fruit and wild mammals (mainly rodents and lagomorphs). Coyotes had higher trophic diversity in their annual diet ($H' = 2.33$) than gray foxes ($H' = 1.80$). When analyzing diets by season, trophic diversity of both species was higher in winter and spring and tended to decrease in summer and autumn. When comparing between species, this parameter differed significantly during all seasons except for summer. Trophic overlap throughout the year was high ($R_O = 0.934$), with seasonal variation between $R_O = 0.821$ (autumn) and $R_O = 0.945$ (spring). Both species based their diet on the most available food items throughout each season of the year, having high dietary overlap which likely can lead to intense exploitative competition processes. However, differences in trophic diversity caused by differential prey use can mitigate competitive interactions, allowing these different sized canid species to coexist in the study area.

**Funding:** CRRL received funding from CONACYT through the doctoral grant #293353; CONACYT as funder, had no role in study design, data collection and analysis, decision to publish, or preparation of the manuscript.

**Competing interests:** The authors have declared that no competing interests exist.

## Introduction

The trophic dimension of species' ecological niches is important because it can determine the structure of ecological communities due to the importance of food resources for animals [1]. Therefore, understanding the ways in which species partition these resource contributes to the understanding of interactions among sympatric species [2, 3]. When species occur in sympatry, the competitive exclusion principle [4] proposes that the species segregate their ecological niches in at least one of their dimensions in order to reduce interspecific competition [5–7]. Differences in size and physiological needs can allow predator species to coexist in the same area [8]. One of the most important forms of resource partitioning in ecological communities is differentiation of the use of food resources [1, 6]. Some degree of trophic overlap is relatively common, and varies among species, sites, and season [9], but cases of very high trophic overlap between ecologically similar species are limited [10]. Comparing the food habits of sympatric species reveals the overlap degree in their trophic niche, which can be interpreted as a measure of the potential for interspecific competition between species [11, 12], and thus provides information of the mechanisms that reduce their competitive interactions in order to maintain sympatry [13]. Also, it is important to state that sympatry can be achieved not only by trophic niche segregation, but also through segregation on other dimensions of ecological niche such as time and space [5–7, 14]. The relevance of niche segregation on each of its main dimensions (e.g. trophic, time and space) for allowing the coexistence of ecologically similar species varies among species and habitats but exploring the overlap on any of those niche dimensions, can provide useful insights on this topic [6]. Although there is abundant evidence of these kind of complex interactions among carnivorous mammals, these interactions are poorly understood for many species as well as the sort of niche segregation they present on any of the ecological niche dimensions [15, 16]. Therefore, we decided to explore the potential role that feeding ecology and trophic interactions among sympatric species of mammalian carnivores can have on their coexistence through trophic niche segregation. To explore these questions we chose studying coyotes (*Canis latrans*) and gray foxes (*Urocyon cinereoargenteus*), two common and widespread canid species, that are abundant at our study site and, sympatric over a large part of their range in North America [17, 18]; however, the ecological relations between them have received little attention [11]. These two canids are generalist-opportunist species that consume similar food items [17, 18] and they potentially compete for similar resources. In the northern portion of their distribution range, the diet overlap of these canids can vary between medium [19, 20] to high values [11, 21, 22]. But in the southern portion of its distribution range, trophic dynamic and dietary overlap between coyotes and gray foxes have not been thoroughly studied. In a tropical dry forest in southern Mexico, it was reported that they showed intermediate trophic niche overlap and low potential for interspecific competition between them [23]; while in the north of the country, in a temperate forest, they showed intermediate–high dietary overlap proportion [24]. As trophic level mechanisms that allow the coexistence of these two canids remain unclear, a study of food resource partitioning in sympatric coyote and gray fox populations in temperate forests will help to understand their competitive relations and to elucidate if trophic niche segregation could be a mechanism partly explaining a stable coexistence between these two canids. Our objective was thus to analyze the trophic interactions and evaluate the potential for interspecific competition for food resources between coyotes and gray foxes in a temperate forest of the Sierra Madre Occidental, in a protected reserve, in the state of Durango, Mexico.

We analyzed the indigestible contents of feces of both species to determine their diet composition, the relative importance of different food items, and its seasonal variation. Using those data, we evaluated: (1) the trophic diversity of these two species, (2) whether this variable

differed between the species, and (3) the similarity of diets as a measure of trophic niche overlap, considering a high overlap degree as an indicator of high potential for exploitative competition [13, 25]. We expected significant differences in trophic diversity between both species, and therefore, low trophic niche overlap. The larger coyote (up to 16 kg in the study area [26]) should consume a wider range of food resources, increasing its trophic niche breadth, in comparison to the smaller gray fox (3–5 kg [27]), as has been reported for this species in other parts of its geographic distribution range.

## Materials and methods

### Study area

We carried out this work in the buffer zone of "La Michilía" Biosphere Reserve (MBR), in the municipality of Súchil, Durango, Mexico, located between the coordinates 23˚ 21'–23˚ 28' N and 104˚ 09'–104˚ 21' W. The MBR is found in the transition zone between the Nearctic and Neotropical biogeographic zones [25–29]. The MBR is bordered by the Sierra de Urica to the west, which is gently sloped, and by the Sierra de Michis on the east, which has steep depressions and marked slopes [30]. The altitude of the study zone varies between 2,000–2,985 m [31]. In the northern part of the MBR the climate is semi-dry temperate (BS1k) and in the rest of the MBR the dominant climate is temperate sub-humid (Cw [32]). Mean monthly temperatures range from 2˚C in February to 22˚C in July and the average annual precipitation ranged from 600–900 mm [32].

Dominant vegetation within the MBR is coniferous (*Pinus* spp.) and oak (*Quercus* spp.) forest, though there are also zones of natural grassland (*Bouteloua* spp.), xerophytic scrub (*Arctostaphylos pungens*, *Acacia schaffneri*), and aquatic vegetation; in addition, the MBR has transition zones among these vegetation types, leading to the formation of mixed forests [33] (Fig 1).

Fig 1 was prepared by the first author (CRRL), for illustrative purposes only, to show location of the study area and vegetation types, using shape files about topography and vegetation types and land cover produced by Mexican National Institute of Geography and Statistics and publicly available for free, for any user at the following link (https://www.inegi.org.mx/datos/?t=0150). Shape files were projected to produce Fig 1 using QGIS software (v. 3.14.) that is free online to download at the following link (https://www.qgis.org/es/site/forusers/download.html).

### Sample collection and identification

During 2018, we selected 23 sampling transects of variable length (between 750–2500 m), to opportunistically collect the feces of both canids along paths, main and secondary roads, and streams in the study area (Fig 1). Based on food-availability cycles in the study area [34], we performed sampling on 17 transects in winter (December 22–March 20) and spring (March 21–June 20), 23 in summer (June 21–September 20) and on 18 in autumn (September 21–December 21). We chose our sampling transects in accordance to the proportion of the main vegetation types present in the study area (ca 124 km², Fig 1), 14 transects on mixed forest (*Pinus* spp.–*Quercus* spp., that represents 62.30% of the study area); 7 transects on pine forest (*Pinus* spp., 28.8% of the study area); 1 transect on oak forest (*Quercus* spp. 4.9% of the study area); and 1 in grasslands (*Bouteloua* spp., 4% of the study area).

Prior to the formal collection of fecal samples, all feces were removed from the transects, allowing us to date the samples to the nearest month during the study period. In the studied area, there are more sympatric carnivores; however, they all produce quite distinctive feces and none can be easily confused with coyote or gray fox feces. Despite this, we were very

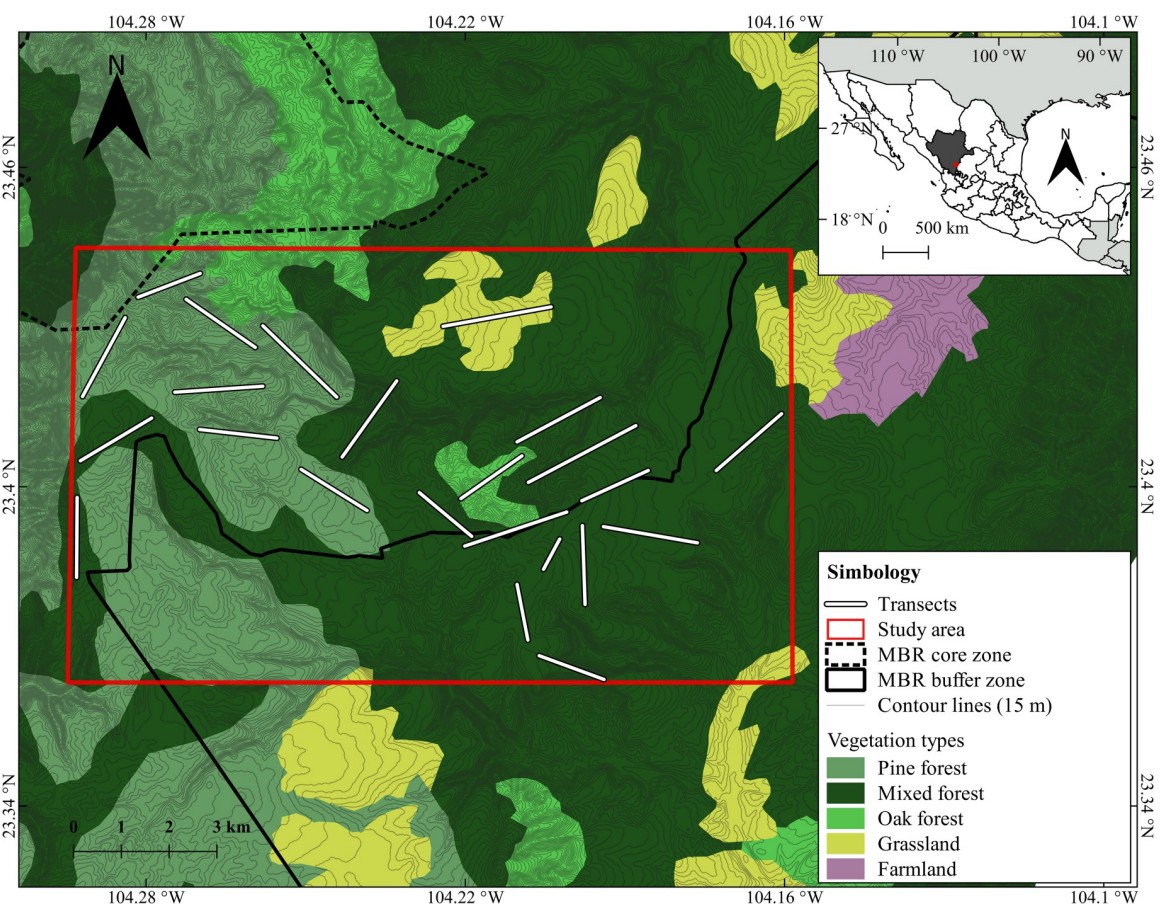

**Fig 1. Geographic location of the study area in the buffer zone of La Michilía Biosphere Reserve (MBR), Durango, Mexico, showing the vegetation types and the location of the sampling transects.** Fig 1 was created by CRRL (the first author) using freely available resources and software (see methods). Therefore, it is an original not copyrighted image.

careful to identify feces to the species level by size, shape, and color of each sample, based on information from specialized literature [35–37], and we considered as complementary evidence the presence of tracks of the focal species in the vicinity of the collection site. Subsequently, to reduce potential assignment error, we used feces maximum diameter as a criterion to identify canid species in the study area, see [34]. Thus, we assigned to *C. latrans* all feces that had maximum diameter between 18.01 and 33.00 mm, and to *U. cinereoargenteus* the feces with maximum diameter between 9.00 and 16.99 mm [35], and we excluded from data analysis all feces with maximum diameter between 17.00 and 18.00 mm and also any disaggregated feces sample. After this, we discarded 7.43% (*n* = 68) of the total feces we collected.

We considered that feces collected on a particular season, were representative of the feeding habits of the studied species in that period of the year. We placed the collected feces individually in paper bags, labeled them with the species name, date, season, and geographic location, to be air dried. In the laboratory, we transferred all feces to nylon stockings and washed them with water and commercial detergent to remove soluble material. We then manually disintegrated the feces and dried them in a 65°C oven for 48 h [34]. After drying, we recovered all the undigested parts of food items (hairs, scales, exoskeletons, bones, skin, teeth, feathers, and seeds); in the case of hairs, we mounted samples of them on microscope slides to visualize their medulla and scale patterns [38]. We then identified the undigested fragments to the

lowest taxonomic level possible using reference samples from the study area from the collection of the Laboratorio de Ecología y Conservación de Fauna Silvestre of the Universidad Autónoma Metropolitana–Unidad Xochimilco, as well as with published information about characteristics of teeth, bones and guarding hairs (mostly qualitative features) of different mammal species from several specialized references [38–44]. Identification of mammal species through their guarding hairs has proven to be reliable, particularly if based on qualitative features [45]. We classified the undigested fragments (i.e., food items) from both canids into six trophic categories: 1) fruits, 2) invertebrates, 3) reptiles, 4) birds, 5) livestock (caprine, ovine, and bovine), and 6) wild mammals. In the case of herbs and/or grass ($n = 4$), we assumed that they were incidentally ingested while consuming small prey or to aid in the digestive process [11], so we excluded them from subsequent analyses.

## Sample analysis

For each species, the overall and seasonal representation of each food item and food category was expressed as: 1) number of occurrences ($n$), $n_i$ = number of feces containing prey item $i$; and 2) frequency of occurrence ($FO$), $FO_i$ (%) = ($n_i / N$) * 100, where $N$ is the total number of feces [21, 46]. The $FO$ measures the percentage of feces that contains a given prey item, and although it does not necessarily approximate the volumetric importance of items in the diet, it indicates the relative importance role of items in the diet [24, 47], how common an item was in the diet [48], and can provide valuable insight into carnivore ecology [21, 49].

We used Clench's asymptotic species accumulation model to estimate the completeness of the sampling. To do this, the data of food items found in feces were randomized 1000 times with the program *EstimateS* version 9.1.0 [50]. Additionally, we used Fisher's exact test [51] to analyze whether the distribution of prey items among the trophic categories varied between species and between seasons. We also calculated Shannon's diversity index ($H'$ [52]) to estimate the trophic diversity, and to identify possible significant differences in trophic diversity between species and among seasons using a Hutcheson's *t* test [53], with the program Past 4.03 [54]. To estimate the trophic niche overlap we used Horn's index ($R_0$ [55]), corrected to avoid bias due to under sampling [56]. The index values range from 0 (no overlap) to 1 (complete overlap). For most of the statistical analysis we used R version 4.0.2 program [57].

Since our data is based on the collection and analysis of feces of the studied species, in Mexico there is no need to obtain a permit for this. We do not collect and handle individuals of the studied species, therefore, we did not have to adhere to a particular ethic guideline for handling and studying animals. To be able to do our research activities inside La Michilía Biosphere Reserve we inform the authorities about the objectives and needs of our research before starting our field work. We were acknowledged that they received our research protocol and that they did not have any issues in letting us do our research inside the protected area.

## Results

We accumulated nearly 28 km of transect sampling effort for winter and spring, almost 30 for autumn and 37 km for winter. We were able to collect a total of 915 fecal samples, but only analyzed 847, 540 from coyotes and 307 from gray foxes. More than 90% of all feces samples were collected as expected, for both species, on mixed and pine forest where the majority of the sampling transects were located. However, we ran a Chi-square goodness of fit test and noticed that we collected significantly less feces samples than expected (considering the proportion of vegetation types on the study area) for both species on grasslands and more than expected on oak forest for coyote and in pine forest for gray fox ($\chi^2 = 9.58$, df = 3, p = 0.022, for coyote; $\chi^2 = 63.64$, df = 3, p < 0.001, for gray fox), suggesting that they use habitats

differently. We identified 25 different food items for the coyote and 17 for the gray fox. We explored how well our sample represents the diet for both species through the Clench model, that predicted 27.16 food items for coyote ($R^2$ = 0.998) and 18.76 food items for gray fox ($R^2$ = 0.978). Thus, we reached 92.05% and 90.62% of the total food items expected, for coyote and gray fox, respectively.

## Overall diet composition

Fisher's exact test indicated that prey items distribution across trophic categories differed significantly (p < 0.001) between coyote and gray fox feces. In the case of coyotes, we identified 25 different food items from all six trophic categories. The category with the highest frequency of occurrence value was fruits ($FO_f$ = 68.52), followed by wild mammals ($FO_{wm}$ = 47.59%) and invertebrates ($FO_i$ = 6.48; Table 1). The fruits category was represented by cedar fruits (*Juniperus deppeana*), which was the predominant food item in coyote feces ($FO_{Jd}$ = 39.07), followed by fruits of pointleaf manzanita (*A. pungens*; $FO_{Ap}$ = 29.44). Among wild mammals category, rodents were the most frequent food item, represented by seven species; the most frequently consumed were mice of the genus *Peromyscus* ($FO_P$ = 10.74) and *Sigmodon* ($FO_S$ = 8.52). In the same category, the next most frequent food items were from the orders Lagomorpha, represented by *Sylvilagus audubonii* and *Lepus californicus* from Leporidae family ($FO_L$ = 8.52); and Artiodactyla, represented by *Odocoileus virginianus* ($FO_{Ov}$ = 3.15), *Pecari tajacu* ($FO_{Pt}$ = 2.04) and the exotic species *Sus scrofa* ($FO_{Ss}$ = 1.67). Among invertebrates, the most frequent food item was of the order Coleoptera ($FO_C$ = 5.00), while, among birds, was *Meleagris gallopavo* ($FO_{Mg}$ = 2.41). In the livestock category, we found remnants of cattle (*Bos taurus*), goats (*Capra hircus*), and sheep (*Ovis aries*), all with low $FO$ values $\leq$ 1.11 (Table 1) and we consider that it might come from the consumption of dead animals more than representative of predation on domestic animals.

For gray foxes, we identified 17 different food items, belonging to 5 of the 6 trophic categories considered. The category with the highest frequency of occurrence value was fruits ($FO_f$ = 84.36), followed by wild mammals ($FO_{wm}$ = 28.66) and invertebrates ($FO_i$ = 9.45; Table 1). We did not find any traces of livestock in gray fox feces. Within fruits category, *J. deppeanna* was the most frequently consumed food item ($FO_{Jd}$ = 54.72) followed by *A. pungens* ($FO_{Ap}$ = 29.64). Within the wild mammals category, rodents were the most frequent food item, represented by six species, among which the most consumed were of the genus *Peromyscus* ($FO_P$ = 11.73), followed by species of the order Lagomorpha ($FO_L$ = 5.21). Among the most frequent invertebrate food item were Coleoptera ($FO_C$ = 7.82), while among birds, the most frequent prey items corresponded to unidentified species ($FO_{NA}$ = 3.58; Table 1).

Trophic diversity was significantly higher (*t* = 7.03, df = 814.22, p < 0.001) for coyotes (*H'* = 2.33) than for gray foxes (*H'* = 1.80). While Horn index was $R_0$ = 0.934 (CI 95%; 0.898–0.969), which indicates high dietary overlap proportion, as well as very similar resource use spectra, between these two canid species.

## Seasonal diet composition

Fisher's exact test showed significant differences in food items distribution across the trophic categories in species feces between seasons for coyotes (p < 0.001) and for gray foxes (p < 0.001). In the case of coyotes, food items in the wild mammals category were the most frequent remains in feces during winter ($FO_{wm}$ = 60.32) and spring ($FO_{wm}$ = 63.08), the first half of the year; while during the second half of the year were food items in the fruits category: summer ($FO_f$ = 84.10) and autumn ($FO_f$ = 66.67). Frequency of occurrence of birds ($FO_b$: 3.59–5.13) and invertebrates ($FO_i$: 4.17–7.18) was relatively constant throughout the year, and

**Table 1. Total number of samples (*N*) and trophic diversity (*H'*) of the coyote and gray fox, as well as overall number of occurrences (*n_i*) and frequency of occurrence (*FO%*) of food items by trophic category, in both canids diets, in the buffer zone of La Michilia Biosphere Reserve (MBR), Durango, Mexico.**

| Food item and trophic category | Coyote $N$ = 540 $H'$ = 2.33 | | Gray fox $N$ = 307 $H'$ = 1.80 | |
| --- | --- | --- | --- | --- |
| | $n_i$ | *FO%* | $n_i$ | *FO%* |
| **Wild mammals category** | **257** | **47.59** | **88** | **28.66** |
| CARNIVORA | | | | |
| Mephitidae | 12 | 2.22 | 1 | 0.33 |
| Procyonidae | | | | |
| *Nasua narica* | 1 | 0.19 | 2 | 0.65 |
| *Procyon lotor* | 1 | 0.19 | 0 | 0 |
| ARTIODACTYLA | | | | |
| Cervidae | | | | |
| *Odocoileus virginianus* | 17 | 3.15 | 0 | 0 |
| Tayassuidae | | | | |
| *Pecari tajacu* | 11 | 2.04 | 0 | 0 |
| Suidae | | | | |
| *Sus scrofa* | 9 | 1.67 | 0 | 0 |
| RODENTIA | | | | |
| Sciuridae | | | | |
| *Sciurus nayaritensis* | 10 | 1.85 | 3 | 0.98 |
| *Otospermophilus variegatus* | 7 | 1.30 | 2 | 0.65 |
| Geomyidae | | | | |
| *Thomomys umbrinus* | 7 | 1.30 | 3 | 0.98 |
| Heteromyidae | | | | |
| *Heteromys irroratus* | 5 | 0.93 | 0 | 0 |
| Cricetidae | | | | |
| *Peromyscus sp.* | 58 | 10.74 | 36 | 11.73 |
| *Reithrodontomys sp.* | 1 | 0.19 | 3 | 0.98 |
| *Sigmodon sp.* | 46 | 8.52 | 19 | 6.19 |
| LAGOMORPHA | | | | |
| Leporidae | 46 | 8.52 | 16 | 5.21 |
| UNIDENTIFIED | 26 | 4.81 | 3 | 0.98 |
| **Livestock category** | **12** | **2.22** | **0** | **0** |
| ARTIODACTYLA | | | | |
| Bovidae | | | | |
| *Bos taurus* | 6 | 1.11 | 0 | 0 |
| *Capra hircus* | 4 | 0.74 | 0 | 0 |
| *Ovis aries* | 2 | 0.37 | 0 | 0 |
| **Birds category** | **24** | **4.44** | **12** | **3.91** |
| GALLIFORMES | | | | |
| Phasianidae | | | | |
| *Meleagris gallopavo* | 13 | 2.41 | 1 | 0.33 |
| Unidentified | 11 | 2.04 | 11 | 3.58 |
| **Reptiles category** | **10** | **1.85** | **3** | **0.98** |
| **Invertebrates category** | **35** | **6.48** | **29** | **9.45** |
| COLEOPTERA | 27 | 5.00 | 24 | 7.82 |
| ORTHOPTERA | 8 | 1.48 | 5 | 1.63 |

*(Continued)*

**Table 1.**  (Continued)

| | Coyote | | Gray fox | |
|---|---|---|---|---|
| | N = 540 | | N = 307 | |
| | H' = 2.33 | | H' = 1.80 | |
| **Fruits category** | **370** | **68.52** | **259** | **84.36** |
| ERICALES | | | | |
| Ericaceae | | | | |
| *Arctostaphylos pungens* | 159 | 29.44 | 91 | 29.64 |
| PINALES | | | | |
| Cupressaceae | | | | |
| *Juniperus deppeana* | 211 | 39.07 | 168 | 54.72 |

reptiles were present only in winter ($FO_r$ = 1.59) and spring ($FO_r$ = 4.10). Livestock were consumed in all seasons except for autumn, with low values between $FO_d$ = 1.54 in summer and $FO_d$ = 3.08 in spring (Fig 2, Table 2).

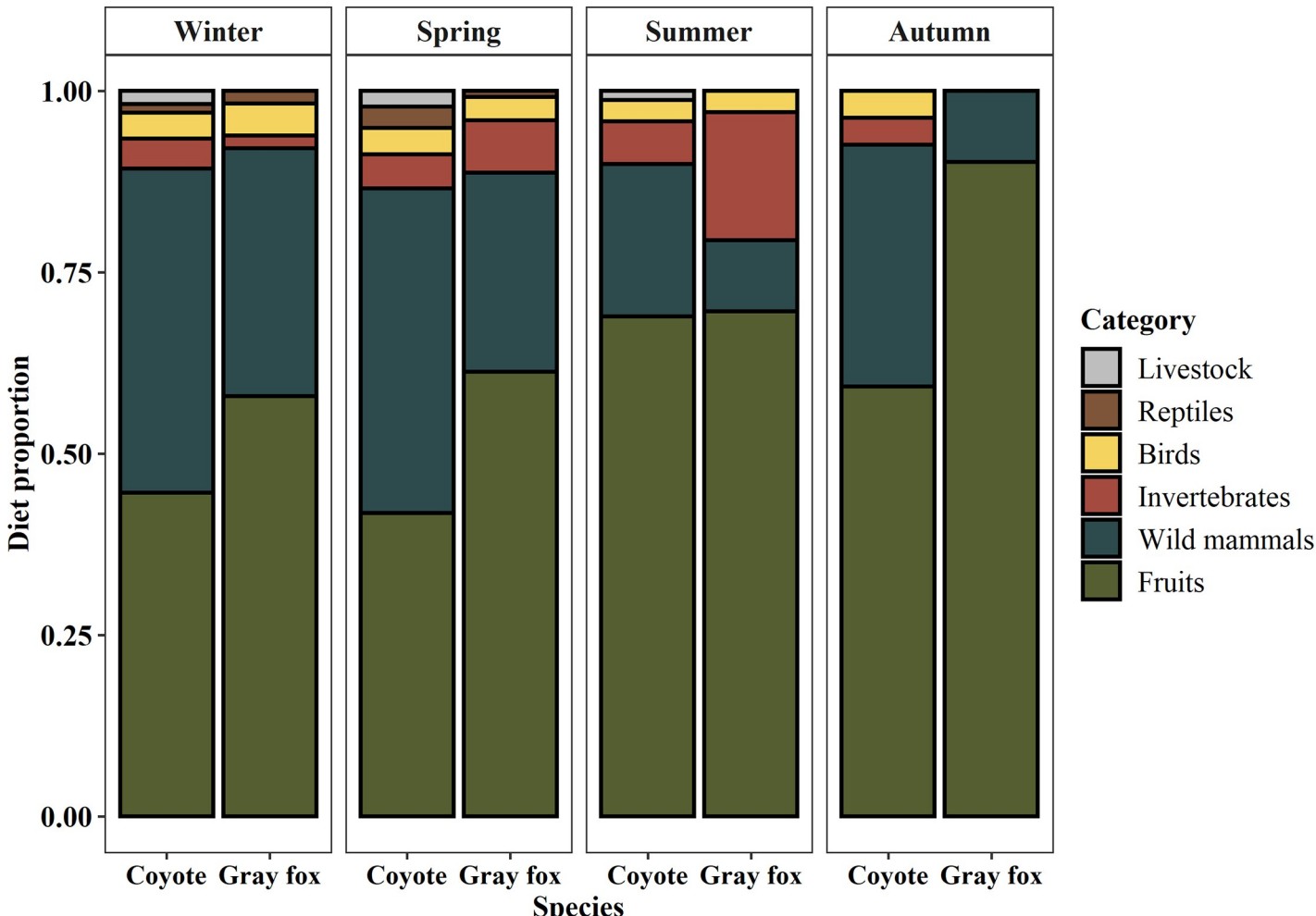

**Fig 2. Seasonal occurrence of food items (expressed as proportion), by trophic categories, in the coyote and gray fox diets in the buffer zone of the La Michilía Biosphere Reserve, Durango, Mexico.**

Gray foxes showed a different pattern of consumption than coyotes, since the most frequent trophic category in all seasons was fruits ($FO_f \geq 75.86$), followed by wild mammals, which were most frequent in winter ($FO_{wm} = 44.83$) and spring ($FO_{wm} = 36.96$). During the summer and autumn, fruits category frequency increased, reaching a maximum value of $FO_f = 92.00$, and frequency of occurrence of wild mammals decreased (Fig 2, Table 2). The highest frequency of occurrence value of invertebrates was during the summer ($FO_i = 23.08$), and no invertebrates were found on its feces in autumn. On the other hand, birds $FO$ values were relatively constant (3.85–5.75), except in autumn when no bird remains were found in the feces. Reptiles were present only in winter ($FO_r = 2.03$) and spring ($FO_r = 1.09$) seasons (Fig 2, Table 2).

The coyote's trophic diversity values were highest in winter and spring ($H' = 2.34$ and $H' = 2.40$, respectively), which did not differ significantly from each other ($t = -0.55$, df = 325.26, p = 0.58), and the minimum values were in summer ($H' = 1.61$) and autumn ($H' = 1.57$), which also did not differ significantly ($t = 0.16$, df = 39.62, p = 0.87). For this species, the remaining pairwise comparisons between seasons shows significant differences using Hutchenson's $t$ tests (Table 3). In the case of gray foxes, like coyotes, the highest trophic diversity values occurred during winter ($H' = 1.97$) and spring ($H' = 1.83$), which did not statistically differ from each other ($t = 1.02$, df = 232.93, p = 0.31), while the lowest value occurred during the autumn ($H' = 0.42$). The rest of the comparisons between seasons showed significant differences (Table 3). When comparing the seasonal trophic diversity values between species, this parameter differed significantly during all seasons except for summer, when there was no evidence of statistically significant differences ($t = 1.27$, df = 267.16, p < 0.205; Table 3).

Trophic niche overlap between coyotes and gray foxes through seasons sampled was high ($\geq 82.10\%$). The highest overlap proportion value occurred in winter with $R_0 = 0.905$ (IC 95%; 0.823–0.986) and spring with $R_0 = 0.945$ (IC 95%; 0.866–0.998), then decreasing in summer with $R_0 = 0.870$ (IC 95%; 0.821–0.919) and showing the lowest overlap in autumn with $R_0 = 0.821$ (IC 95%; 0.673–0.968).

## Discussion

In the study area we found that fruits were the most frequently consumed food item year-round in coyote's diet, as has been reported for this species in coniferous forests habitats in Oregon and Maine [58, 59] and in some desert, coastal, tropical deciduous forest, and urban environments [23, 60–64]. But, this results contrast with previous works in the MBR [24, 34], although former work represents a partial approximation of the diet throughout the year during the summer and spring, and with works in other parts of coyote's distribution which report wild mammals as the most consumed trophic category year-round in temperate forest of Mexico [65–67], the United States of America [13, 68–74], and Canada [75–77]. However, despite the above, changes in the frequency of occurrence of the different trophic categories through the seasons coincide with data reported for this species for which the highest consumption of wild mammals occurred during the first half of the year, in winter and spring, and the main consumption of fruits occurred in the second part of the year, in summer and autumn [34, 64, 68, 74, 78–82]. Our results support that the coyote behaved as opportunistic feeder with general diet [68, 83]. This species is characterized by its adaptability to different habitat conditions, which is reflected in its diet, in such a way that coyotes took advantage of the seasonal availability of mammals and fruits. As was reported in the study area [34], we found that the highest consumption of wild mammals occurred in winter: mainly rodents, lagomorphs, and artiodactyls, which coincides with data reported for this species, since rodents of the genus *Peromyscus* and *Sigmodon* as well as lagomorphs (*L. californicus* and *S.*

**Table 2. Seasonal number of samples (*N*) and trophic diversity (*H'*) of the coyote and gray fox, as well as seasonal number of occurrences (*n_i*) and frequency of occurrence (*FO%*) of food items by trophic category, in both canids diets, in the buffer zone of La Michilia Biosphere Reserve (MBR), Durango, Mexico.**

| Food item and trophic category | Coyote | | | | | | | | Gray fox | | | | | | | |
|---|---|---|---|---|---|---|---|---|---|---|---|---|---|---|---|---|
| | Winter | | Spring | | Summer | | Autumn | | Winter | | Spring | | Summer | | Autumn | |
| | N = 126 | | N = 195 | | N = 195 | | N = 24 | | N = 87 | | N = 92 | | N = 78 | | N = 50 | |
| | H' = 2.16 | | H' = 2.30 | | H' = 1.47 | | H' = 1.57 | | H' = 1.88 | | H' = 1.77 | | H' = 1.36 | | H' = 0.36 | |
| | *n_i* | *FO%* | *n_i* | *FO%* | *n_i* | *FO%* | *n_i* | *FO%* | *n_i* | *FO%* | *n_i* | *FO%* | *n_i* | *FO%* | *n_i* | *FO%* |
| **Wild mammals category** | **76** | **60.32** | **123** | **63.08** | **50** | **25.64** | **9** | **37.50** | **39** | **44.83** | **34** | **36.96** | **10** | **12.82** | **5** | **10.00** |
| Mephitidae | 6 | 4.76 | 3 | 1.54 | 3 | 1.54 | 0 | 0.00 | 1 | 1.15 | 0 | 0.00 | 0 | 0.00 | 0 | 0.00 |
| *N. narica* | 1 | 0.79 | 0 | 0.00 | 0 | 0.00 | 0 | 0.00 | 2 | 2.30 | 0 | 0.00 | 0 | 0.00 | 0 | 0.00 |
| *P. lotor* | 1 | 0.79 | 0 | 0.00 | 0 | 0.00 | 0 | 0.00 | 0 | 0.00 | 0 | 0.00 | 0 | 0.00 | 0 | 0.00 |
| *O. virginianus* | 4 | 3.17 | 11 | 5.64 | 2 | 1.03 | 0 | 0.00 | 0 | 0.00 | 0 | 0.00 | 0 | 0.00 | 0 | 0.00 |
| *P. tajacu* | 2 | 2.38 | 6 | 3.08 | 3 | 1.54 | 0 | 0.00 | 0 | 0.00 | 0 | 0.00 | 0 | 0.00 | 0 | 0.00 |
| *S. scrofa* | 4 | 3.17 | 3 | 1.54 | 2 | 1.03 | 0 | 0.00 | 0 | 0.00 | 0 | 0.00 | 0 | 0.00 | 0 | 0.00 |
| *S. nayaritensis* | 3 | 2.38 | 4 | 2.05 | 3 | 1.54 | 0 | 0.00 | 2 | 2.30 | 1 | 1.09 | 0 | 0.00 | 0 | 0.00 |
| *O. variegatus* | 1 | 0.79 | 2 | 1.03 | 2 | 1.03 | 2 | 8.33 | 1 | 1.15 | 0 | 0.00 | 1 | 1.28 | 0 | 0.00 |
| *T. umbrinus* | 0 | 0.00 | 5 | 2.56 | 2 | 1.03 | 0 | 0.00 | 2 | 2.30 | 1 | 1.09 | 0 | 0.00 | 0 | 0.00 |
| *H. irroratus* | 2 | 1.59 | 0 | 0.00 | 3 | 1.54 | 0 | 0.00 | 0 | 0.00 | 0 | 0.00 | 0 | 0.00 | 0 | 0.00 |
| *Peromyscus* sp. | 22 | 17.46 | 27 | 13.85 | 7 | 3.59 | 2 | 8.33 | 16 | 18.39 | 14 | 15.22 | 4 | 5.13 | 2 | 4.00 |
| *Reithrodontomys* sp. | 0 | 0.00 | 1 | 0.51 | 0 | 0.00 | 0 | 0.00 | 0 | 0.00 | 1 | 1.09 | 2 | 2.56 | 0 | 0.00 |
| *Sigmodon* sp. | 10 | 7.94 | 29 | 14.87 | 6 | 3.08 | 1 | 4.17 | 7 | 8.05 | 7 | 7.61 | 3 | 3.85 | 2 | 4.00 |
| Leporidae | 13 | 10.32 | 19 | 9.74 | 12 | 6.15 | 3 | 12.50 | 7 | 8.05 | 9 | 9.78 | 0 | 0.00 | 0 | 0.00 |
| Unidentified | 7 | 5.56 | 13 | 6.67 | 5 | 2.56 | 1 | 4.17 | 1 | 1.15 | 1 | 1.09 | 0 | 0.00 | 1 | 2.00 |
| **Livestock category** | **3** | **2.38** | **6** | **3.08** | **3** | **1.54** | **0** | **0.00** | **0** | **0.00** | **0** | **0.00** | **0** | **0.00** | **0** | **0.00** |
| *B. taurus* | 2 | 1.59 | 3 | 1.54 | 1 | 0.51 | 0 | 0.00 | 0 | 0.00 | 0 | 0.00 | 0 | 0.00 | 0 | 0.00 |
| *C. hircus* | 0 | 0.00 | 2 | 1.03 | 2 | 1.03 | 0 | 0.00 | 0 | 0.00 | 0 | 0.00 | 0 | 0.00 | 0 | 0.00 |
| *O. aries* | 1 | 0.79 | 1 | 0.51 | 0 | 0.00 | 0 | 0.00 | 0 | 0.00 | 0 | 0.00 | 0 | 0.00 | 0 | 0.00 |
| **Birds category** | **6** | **4.76** | **10** | **5.13** | **7** | **3.59** | **1** | **4.17** | **5** | **5.75** | **4** | **4.35** | **3** | **3.85** | **0** | **0.00** |
| *M. gallopavo* | 3 | 2.38 | 7 | 3.59 | 3 | 1.54 | 0 | 0.00 | 1 | 1.15 | 0 | 0.00 | 0 | 0.00 | 0 | 0.00 |
| Unidentified | 3 | 2.38 | 3 | 1.54 | 4 | 2.05 | 1 | 4.17 | 4 | 4.60 | 4 | 4.35 | 3 | 3.85 | 0 | 0.00 |
| **Reptiles category** | **2** | **1.59** | **8** | **4.10** | **0** | **0.00** | **0** | **0.00** | **2** | **2.30** | **1** | **1.09** | **0** | **0.00** | **0** | **0.00** |
| **Invertebrates category** | **7** | **5.56** | **13** | **6.67** | **14** | **7.18** | **1** | **4.17** | **0** | **0.00** | **9** | **9.78** | **18** | **23.08** | **0** | **0.00** |
| Scarabaeidae | 3 | 2.38 | 12 | 6.15 | 11 | 5.64 | 1 | 4.17 | 2 | 2.30 | 8 | 8.70 | 14 | 17.95 | 0 | 0.00 |
| Orthoptera | 4 | 3.17 | 1 | 0.51 | 3 | 1.54 | 0 | 0.00 | 0 | 0.00 | 1 | 1.09 | 4 | 5.13 | 0 | 0.00 |
| **Fruits category** | **75** | **59.52** | **115** | **58.97** | **164** | **84.10** | **16** | **66.67** | **66** | **75.86** | **76** | **82.61** | **71** | **91.03** | **46** | **92.00** |
| *A. pungens* | 61 | 48.41 | 87 | 44.62 | 10 | 5.13 | 1 | 4.17 | 35 | 40.23 | 45 | 48.91 | 11 | 14.10 | 0 | 0.00 |
| *J. deppeana* | 14 | 11.11 | 28 | 14.36 | 154 | 78.97 | 15 | 62.50 | 31 | 35.63 | 31 | 33.70 | 60 | 76.92 | 46 | 92.00 |

*audubonii*) were the elements that accounted for the highest proportion of consumption [11, 13, 24, 34, 65]. This species has also been reported to consume larger species, such as white-tailed deer (*O. virginianus*), which in the case of the present study had high *FO* values, below rodents and lagomorphs, but which may present higher importance values in other distribution areas [66, 68, 74, 76, 84–88]. These high values of mammal consumption during the first half of the year can be explained by the high demand for high-quality foods, since these periods coincide with the breeding season (1 January–15 March) and gestation (16 March–30 April [89]), such that foraging activity increases markedly in the study area to increase reproductive success [26, 90]. On the other hand, in summer and autumn, fruit frequency was higher; the most consumed fruits were from cedar (*J. deppeana*), a species which has high availability during this season, July-November [34], while the main consumption of *A. pungens* was during

**Table 3. Seasonal trophic diversity of coyotes and gray foxes, and significance values of Hutcheson's *t* test between seasons, in the buffer zone of La Michilía Biosphere Reserve (MBR), Durango, Mexico.**

| | | | Coyote | | | | Gray fox | | | |
|---|---|---|---|---|---|---|---|---|---|---|
| | | | Winter | Spring | Summer | Autumn | Winter | Spring | Summer | Autumn |
| | | | $H' = 2.16$ | $H' = 2.30$ | $H' = 1.47$ | $H' = 1.57$ | $H' = 1.88$ | $H' = 1.77$ | $H' = 1.36$ | $H' = 0.36$ |
| Coyote | Winter | $H' = 2.16$ | ---- | | | | | | | |
| | Spring | $H' = 2.30$ | NS | ---- | | | | | | |
| | Summer | $H' = 1.47$ | *** | *** | ---- | | | | | |
| | Autumn | $H' = 1.57$ | ** | *** | NS | ---- | | | | |
| Gray fox | Winter | $H' = 1.88$ | ** | *** | ** | NS | ---- | | | |
| | Spring | $H' = 1.77$ | *** | *** | NS | NS | NS | ---- | | |
| | Summer | $H' = 1.36$ | *** | *** | NS | NS | *** | ** | ---- | |
| | Autumn | $H' = 0.36$ | *** | *** | *** | *** | *** | *** | *** | ---- |

Minimum number of samples between comparisons was 74, degrees of freedom varied between 37.71–407.10. Significance values are indicated as follows: $P \leq 0.001$ "***", $P \leq 0.01$ "**", $P \leq 0.05$ "*", and $P > 0.05$ "NS".

the winter and spring, which coincides with the low-water period in this area, just when the fruits of this shrub mature and fall to the ground; thus, coyotes consume the fruit and help disperse the seeds of this species which has an important role in this type of ecosystem, especially in restoring soils and retaining moisture at the beginning of secondary ecological succession [91]. Although our work shows that the rest of the food categories were complementary elements in the coyote diet, their importance and frequency of consumption vary depending on the habitat type where this canid resides, since in desert regions there is a higher consumption of invertebrates and reptiles [92, 93] and in anthropized environments, an important consumption of livestock ($FO \geq 25\%$), including cattle, poultry, and domestic cats has been reported [94–97]. It is cautionary to state that domestic animal remnants detected in coyote´s feces is likely represent carrion consumption more than predation events. In fact, in the study area there are few reports of livestock predation events from local people, and none reported in the study year of our work.

Overall feeding habits of the gray fox that we describe in this work, in which the main trophic category was fruits followed by wild mammals, coincide with the feeding patterns reported in similar biomes in North America [20, 70, 98, 99], in central Mexico [100], in Guatemala [101], and in some areas of moist tropical forest in Mexico [23, 102] and Belize [103]. The gray foxes' most consumed trophic category throughout the year was fruits. However, the frequency of consumption of this and the rest of the categories differed significantly among seasons. During winter there was higher consumption of wild mammals, as has been reported in the central and eastern USA, where leporids and rodents are the gray fox's main prey [18, 104]. The lowest consumption of fruits occurred in the winter and increased gradually through the autumn, when it represented more than 90% of the diet. During autumn, foxes consumed almost exclusively cedar (*J. depeanna*) fruits. Invertebrates was also highly consumed by gray foxes, which have been reported to be mainly consumed in the summer [18, 19, 23, 105–107], the wettest season, as was the case in our study.

The variation in the frequency of consumption in the trophic categories that make up the diets of these two canids shows their ability to adapt to different habitat conditions. In the case of the coyote, it has been reported that this species responds to changes in resource availability by modifying their preferences when an important food source becomes less abundant [20, 34, 108, 109]. This also appears to occur with gray foxes, since they have also demonstrated their

adaptability to changes in the availability of food resources, whether due to stochastic events, temporal variation, or differences in the habitat types they occupy [18, 20, 23, 24, 102].

As expected, yearly trophic diversity of the coyote ($H' = 2.33$) was higher than that of the gray fox ($H' = 1.80$). Seasonally, this parameter differed between the two species in all seasons except for the summer, when the diversity of the dietary elements of the two species was more similar (82.15%). Coyotes had its highest trophic diversity in spring ($H' = 2.40$), and this was higher than that of the gray foxes ($H' = 1.97$). During this season, the coyote consumed three different species of livestock (*B. taurus*, *O. aries* and *C. hircus*) and three species of large wild mammals (*O. virginianus*, *P. tajacu* and the exotic *S. scrofa*), which the gray foxes did not consume at any time during the year. This is consistent with the prediction that sympatric carnivore species will partition prey species according to their body size [8], which may reflect the different energy requirements associated with size [13]. This has been previously reported in the coyote with respect to the gray fox [11, 24] and to other species of foxes in the Americas, such as the San Joaquín kit fox (*Vulpes macrotis mutica* [110, 111]), the swift fox (*Vulpes velox* [13]), and the red fox (*Vulpes vulpes* [112, 113]).

We expected that both species should have low trophic niche overlap in the area, however, despite the differences we found in trophic diversity between these two species, trophic niche overlap between coyotes and gray foxes in the study area was high overall (93.4%) and seasonally (82.1–94.5%), since both species consumed many of the same food items that were available in the MBR; this suggests that there could be some level of exploitative competition. However, the relevance of each trophic category differed between species. The *FO* of wild mammals was 1.6 times higher in the coyote's than in the gray fox diet. Invertebrates and fruit presented *FO* values, 1.45 and 1.23 times higher, respectively, in the gray fox than in the coyote diet. Also, livestock was consumed only by the coyote. Taken together, this suggests that even when the overlap value of the diet is high, each species consumes with different emphasis some of these shared resources, which may mitigate potential competition for food resources [8].

Our results coincided with the only previous study investigating the similarity of the diets of these two sympatric canids in the study area, that reported high trophic overlap, with a Pianka index value of $O = 0.832$ [24]. Given that the abundance of potential prey should be similar for both predator species, some of the differences detected in the use of prey could reflect differences in the feeding ecology of these two species [24], so competitive interactions over food resources are mediated by distinct foraging patterns that result in a differentiation of the consumption of some of the elements of the diets of these species. This resource partitioning pattern by ecologically similar species is a niche segregation strategy that can facilitates the coexistence between them [5, 6, 19, 70]. However, it should also be pointed out that this difference in consumption patterns could also be due to differential use of the habitat (e. g. spatial segregation), which has been shown to be important in reducing competitive interactions between coyotes and gray foxes [19, 70, 114]. We collected more or less fecal samples by vegetation type than expected by its proportion on the study area, however we consider that not much inference about food resources consumed by habitat type can be derived from the location site of the feces sample, since an animal can consume resources at one place, and defecate several hours later in a different habitat type. However, t differences among habitats in the success to collect feces samples, indeed can provide some evidence of differential use of vegetation types, something that is in accordance with our findings about spatial ecology and habitat use data obtained at the same site [114]. Additionally, different behavioral strategies can also be employed by species to facilitate coexistence with a dominant species [115]. In the case of interactions between the coyote and the gray fox, it has been shown that the strategies employed by subdominant species include spatial [19, 116] and temporal [116] predator avoidance.

Both canid species showed opportunistic food habits, although the trophic diversity of the coyote was higher than that of the gray fox, mainly because coyote incorporated larger prey into its diet than gray fox did not. However, the two species consumed quite similar range of food items, that varied in proportion and frequency among seasons, so they showed high trophic niche overlap, and therefore some potential for exploitative competition in the study area. For a better understanding on how the different foraging strategies of each species can help minimize their overlap in the trophic niche and facilitate their coexistence, more detailed research is needed on the availability and spatio-temporal dispersion of food resources in the different habitat types in the area. Such information could also provide insights on how the implementation of different management actions of habitats and/or species can affect both canid species persistence in this Natural Protected Area.

## Supporting information

**S1 Dataset. Undigested remains 2018.** Undigested remains from 2018 coyote and gray fox feces.
(XLSX)

**S1 File. Resumen.**
(DOCX)

## Acknowledgments

We thank the authorities of La Michilía Biosphere Reserve, Ejido San Juan de Michis and Anexo La Peña, Durango, Mexico, for the permits granted to carry out this work. To INECOL A. C. for allowing us to use the installations of "Piedra Herrada" Biological Station and to J. Medina and A. Guerra from that institution for their support. We are grateful to M. Villa, P. Villa, E. Chacón and D. Carreón for logistical and/or field work support. We also thank M. Sánchez for laboratory work support in the identification of hair samples. The Laboratorio de Ecología y Conservación de Fauna Silvestre of the UAM-X, VISILMEX A. C. and the Centro de Investigación en Biodiversidad y Conservación of the UAEM provided infrastructure, supplies, and all-terrain vehicles. This article was the result of the dissertation work of the first author to obtain a doctoral degree in the Doctorado en Ciencias Biológicas y de la Salud at the Universidad Autónoma Metropolitana.

## Author Contributions

**Conceptualization:** César Ricardo Rodríguez-Luna, Jorge Servín, David Valenzuela-Galván, Rurik List.

**Data curation:** César Ricardo Rodríguez-Luna.

**Formal analysis:** César Ricardo Rodríguez-Luna.

**Investigation:** César Ricardo Rodríguez-Luna.

**Methodology:** César Ricardo Rodríguez-Luna, Jorge Servín, David Valenzuela-Galván, Rurik List.

**Resources:** César Ricardo Rodríguez-Luna, Jorge Servín.

**Supervision:** Jorge Servín, David Valenzuela-Galván.

**Validation:** César Ricardo Rodríguez-Luna.

**Visualization:** César Ricardo Rodríguez-Luna.

**Writing – original draft:** César Ricardo Rodríguez-Luna, Jorge Servín, David Valenzuela-Galván, Rurik List.

**Writing – review & editing:** César Ricardo Rodríguez-Luna, David Valenzuela-Galván.

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
