## [Decision Letter · Decision Letter 0]

9 Aug 2021

PONE-D-21-19581

Trophic niche overlap between coyotes and gray foxes in a temperate forest in Durango, Mexico

PLOS ONE

Dear Dr. Valenzuela-Galván,

Thank you for submitting your manuscript to PLOS ONE. After careful consideration, we feel that it has merit but does not fully meet PLOS ONE’s publication criteria as it currently stands. Therefore, we invite you to submit a revised version of the manuscript that addresses the points raised during the review process.

The manuscript is well written and the data presented are valuable. However, several parts can be improved following the reviewers' suggestions (see also the attached file). Particular care must be given to the introduction and conclusion (discuss mechanisms allowing niche overlap and coexistence) and the methods (more details on the transects and fecal collection). 

We look forward to receiving your revised manuscript.

Kind regards,

Nicoletta Righini, PhD

Academic Editor

PLOS ONE

Journal Requirements: 

4.  We note that Figure (1) in your submission contain map images which may be copyrighted. All PLOS content is published under the Creative Commons Attribution License (CC BY 4.0), which means that the manuscript, images, and Supporting Information files will be freely available online, and any third party is permitted to access, download, copy, distribute, and use these materials in any way, even commercially, with proper attribution. For these reasons, we cannot publish previously copyrighted maps or satellite images created using proprietary data, such as Google software (Google Maps, Street View, and Earth). For more information, see our copyright guidelines: http://journals.plos.org/plosone/s/licenses-and-copyright.

1. You may seek permission from the original copyright holder of Figure (1) to publish the content specifically under the CC BY 4.0 license.  

Reviewers' comments:

Reviewer's Responses to Questions

**Comments to the Author**

1. Is the manuscript technically sound, and do the data support the conclusions?

Reviewer #1: Partly

Reviewer #2: No

2. Has the statistical analysis been performed appropriately and rigorously? 

Reviewer #1: Yes

Reviewer #2: Yes

3. Have the authors made all data underlying the findings in their manuscript fully available?

Reviewer #1: No

Reviewer #2: No

4. Is the manuscript presented in an intelligible fashion and written in standard English?

Reviewer #1: Yes

Reviewer #2: Yes

5. Review Comments to the Author

**Reviewer #1:** This manuscript provides a simple, straightforward examination of dietary niche overlap between sympatric coyotes and gray foxes. As the authors point out, there is a lack of data examining the partitioning of these two ecologically similar and highly overlapping carnivores. I believe the authors’ research provides valuable information on the dietary niches of these two species. However, I also note a number of points that could be addressed to strengthen the manuscript.

1. The introduction could benefit from a bit more thoughtful expansion. The authors state that high trophic overlap for ecologically similar species is rare (line 45), but I might disagree. There are many sympatric generalist species, including carnivores (such as foxes, coyotes and bobcats), that can have high trophic overlap but coexist through mechanisms such as fine-scale spatial and/or temporal partitioning. In fact, the authors mention a study where coyotes and gray foxes can be found to have high dietary overlap (line 55). I believe it would strengthen the introduction to discuss the mechanisms which allow sympatric species with overlapping trophic niches to have stable coexistence. The authors mention that trophic niche segregation may allow stable coexistence (line 64), but what about circumstances (such as the findings in this manuscript) where trophic niche overlap is high but coexistence still appears to occur?

2. The transition from the broad introduction to the authors’ specific research seems a bit abrupt. Further discussion on how the reasoning and application of this research may be prudent.

3. In the Sample Collection and Identification section, a few points may benefit from clarification which could improve future design replication:

a. How were transects allocated with respect to vegetation type? What was the length(s) of each transect?

b. This is quite minor, but it feels inappropriate to say all feces were removed from the study area (line 99). Instead, “all feces were removed from the transects” might be more appropriate.

c. Were there any other sympatric carnivores that could be mis-identified as grey fox or coyote (e.g. Mephitidae species or other Vulpes species)? Especially if so, were all other scats cleared from transects during each survey?

4. Although the authors mention sampling transects in each vegetation type, and that coyote food item consumption may vary by habitat type (line 276), vegetation type is not included in any analysis or discussion. The authors find high dietary overlap, but do not discuss any other mechanisms which may facilitate stable coexistence. Examining the spatial overlap (site based or vegetation type based) of both species occurrences and dietary items may further illuminate the mechanisms which allow for sympatry of these two species. While this is briefly mentioned as a potential mechanism (lines 325 – 326), the authors likely have the necessary data to examine a spatial component as well (location and vegetation association of scat samples) to improve their findings.

5. The authors do a good job of thoroughly reviewing the findings in the discussion, but could strengthen their conclusions with greater connections back to the introduction. The authors do not relate back to their prediction in the introduction (lines 73 – 77), and I believe this would be important especially because the authors found high and not low trophic niche overlap. Due to these findings, further discussion on if/how stable coexistence may be occurring could be beneficial. The final conclusion in the discussion feels a bit lacking and leaves the reader wondering how these findings are applicable and where researchers/managers might benefit and move forward.

6. The authors have generally written a nice manuscript, but it could be improved with some editing to correct grammar and improve sentence flow. A few examples are:

a. Lines 33 – 35: the first “difference” would read better as “differences” ; “these different body sizes canid species” perhaps instead could be “these different sized canid species”

b. Line 47 and 59: what is “them” referencing ?

c. Line 55: “its distribution the overlap of its diets” – what is “its” referencing?

d. Lines 61 – 65 and lines 73 - 77: these sentences are a bit hard to work through and could be improved with grammatical changes and/or being broken into two sentences.

**Reviewer #2:** The subject is interesting. However, it lacks many details of the methodology in the collection of samples. It is important to have information on the period with which the transects traveled, if in all the collections they traveled the same transects. This is due to the fact that generalist and opportunistic species such as the coyote and the gray fox can vary their diet in reduced time and space. On the other hand, the morphological characterization of the excrements would be convenient to strengthen it with some chemical or molecular technique; because its morphology is similar and the probability of error is high. On the other hand, it is important to include complementary material with the characteristics of the hair of the identified species because several are similar.

6. PLOS authors have the option to publish the peer review history of their article (what does this mean?). If published, this will include your full peer review and any attached files.

Reviewer #1: No

Reviewer #2: **Yes: **octavio monroy-vilchis

---

## [Author Response · Author response to Decision Letter 0]

30 Aug 2021

DETAILED DESCRIPTION ON HOW DO WE ATTEND ALL COMMENTS AND SUGGESTIONS DONE BY THE ASSOCIATE EDITOR AND BY TWO REVIEWERS

MANUSCRIPT PONE-S-21-24256 Trophic niche overlap between coyotes and gray foxes in a temperate forest in Durango, Mexico authored by César Ricardo Rodríguez-Luna, Jorge Servín, David Valenzuela-Galván (as corresponding author) and by Rurik List

Requests of the Associate Editor and/or by the journal 

R= We have done the required changes, and double checked that we do follow correctly PLOS ONE’s style requirements in the newer corrected version we are now submitting.

R= Since our data is based on the collection and analysis of feces of the studied species, in México there is no need to obtain a permit for this. We do not collect and handle individuals of the studied species, therefore, we did not have to adhere to a particular ethic guideline for handling and studying animals.

In México, to be able to do research activities inside a Natural Protected Area, it is customary to inform the authorities of the NPA about the objectives and needs of the particular research; we did that before starting our research at La Michilia Biosphere Reserve. Since it is only needed to inform the authorities of the NPA, we only received verbal confirmation that they received our research protocol and that they do not had any issues in let in us to do our research, that provides useful information for the management of natural resources within the NPA.

R= We have now uploaded a raw data base as supporting information that contains all needed data for anyone to perform the same analysis we did and present in our manuscript and to replicate the reported study findings in their entirety.

4. We note that Figure (1) in your submission contain map images which may be copyrighted. All PLOS content is published under the Creative Commons Attribution License (CC BY 4.0), which means that the manuscript, images, and Supporting Information files will be freely available online, and any third party is permitted to access, download, copy, distribute, and use these materials in any way, even commercially, with proper attribution. For these reasons, we cannot publish previously copyrighted maps or satellite images created using proprietary data, such as Google software (Google Maps, Street View, and Earth). For more information, see our copyright guidelines: http://journals.plos.org/plosone/s/licenses-and-copyright.

1. You may seek permission from the original copyright holder of Figure (1) to publish the content specifically under the CC BY 4.0 license. 

R= figure 1 was created by CRRL (the first author) using shape files about topography and vegetation types and land cover produced by INEGI (Mexican National Institute of Geography and Statistics) and publicly available for free, for any user at the following link (https://www.inegi.org.mx/datos/?t=0150). Shape files were projected to produce figure 1 using QGIS software (v. 3.14.) that is free online to download at the following link (https://www.qgis.org/es/site/forusers/download.html). We stated now in our corrected version within the Methods section the source for this information and we stated now at the figure 1 caption that it was done in that way by CRRL and therefore we do not need to seek permission from anyone.

Reviewer's Comments to the Author

Reviewer #1: 

This manuscript provides a simple, straightforward examination of dietary niche overlap between sympatric coyotes and gray foxes. As the authors point out, there is a lack of data examining the partitioning of these two ecologically similar and highly overlapping carnivores. I believe the authors’ research provides valuable information on the dietary niches of these two species. 

R= we thank and appreciate reviewer’s considerations to our manuscript.

However, I also note a number of points that could be addressed to strengthen the manuscript.

1. The introduction could benefit from a bit more thoughtful expansion. The authors state that high trophic overlap for ecologically similar species is rare (line 45), but I might disagree. There are many sympatric generalist species, including carnivores (such as foxes, coyotes and bobcats), that can have high trophic overlap but coexist through mechanisms such as fine-scale spatial and/or temporal partitioning. In fact, the authors mention a study where coyotes and gray foxes can be found to have high dietary overlap (line 55). I believe it would strengthen the introduction to discuss the mechanisms which allow sympatric species with overlapping trophic niches to have stable coexistence. The authors mention that trophic niche segregation may allow stable coexistence (line 64), but what about circumstances (such as the findings in this manuscript) where trophic niche overlap is high but coexistence still appears to occur?

R= we have now added more information and a more thoughtful expansion on the issues mentioned by reviewer 1 to our introduction.

2. The transition from the broad introduction to the authors’ specific research seems a bit abrupt. Further discussion on how the reasoning and application of this research may be prudent.

R= We have added a bit more information about our reasoning of our research and why we consider that might provide insights on how the trophic niche segregation can explain partially the coexistence of sympatric ecologically similar species. Also, we modified the writing to solve the abrupt transition between the broad introduction and our specific research.

3. In the Sample Collection and Identification section, a few points may benefit from clarification which could improve future design replication:

a. How were transects allocated with respect to vegetation type? What was the length(s) of each transect?

R= We have provided now information about the number of transects, its length and location in relation to vegetation.

b. This is quite minor, but it feels inappropriate to say all feces were removed from the study area (line 99). Instead, “all feces were removed from the transects” might be more appropriate.

R= we accept and add this suggestion to our ms.

c. Were there any other sympatric carnivores that could be mis-identified as grey fox or coyote (e.g., Mephitidae species or other Vulpes species)? Especially if so, were all other scats cleared from transects during each survey?

R= In the studied area, there is only two wild canid species, but indeed, there are more sympatric carnivores, however, all produce quite distinctive feces and none can be easily confused with coyote or gray fox feces. In our original version of the manuscript, we stated that we identify all collected feces based on size, color, shape, length and maximum diameter and its comparison with the data for feces of different mammalian carnivore species published on field guides for different areas of the country. We discarded any collected feces that could be assigned to other species different of Coyote or Gray Fox. Asides, in the area, a previous study (Servin and Huxley, 1991) collected and measured different morphological data of a quite big sample of feces of Coyote and Gray Fox. We used the morphological data produced by them, to classify our collected feces as Coyote or Gray Fox feces, discarding all feces with maximum diameter falling in the overlapping area of the diameter distribution range of each species. We have added a bit more detail of this to our corrected version of the manuscript.

4. Although the authors mention sampling transects in each vegetation type, and that coyote food item consumption may vary by habitat type (line 276), vegetation type is not included in any analysis or discussion. 

R= we have now changed our figure one to show the vegetation types and the distribution sampling transects in the studied area. Asides, in the text of the manuscript we briefly mention how many sampling transects were located on each vegetation type and we explain in the manuscript that fecal samples of both species were collected on all vegetation types and sampling was in accordance to the proportion of each habitat type. We added a couple of lines to our discussion stating that although feces were collected in all habitat types, not much inference about food resources consumed by habitat type can be derived from the location site of the feces sample, since an animal can consume resources at one place, and defecate several hours before at a different habitat type.

The authors find high dietary overlap, but do not discuss any other mechanisms which may facilitate stable coexistence. Examining the spatial overlap (site based or vegetation type based) of both species occurrences and dietary items may further illuminate the mechanisms which allow for sympatry of these two species. While this is briefly mentioned as a potential mechanism (lines 325 – 326), the authors likely have the necessary data to examine a spatial component as well (location and vegetation association of scat samples) to improve their findings.

R= We appreciate this reviewer’s observation and we have now expanded our discussion to include information about any other mechanism that can facilitate stable coexistence.

5. The authors do a good job of thoroughly reviewing the findings in the discussion, but could strengthen their conclusions with greater connections back to the introduction. The authors do not relate back to their prediction in the introduction (lines 73 – 77), and I believe this would be important especially because the authors found high and not low trophic niche overlap. Due to these findings, further discussion on if/how stable coexistence may be occurring could be beneficial. The final conclusion in the discussion feels a bit lacking and leaves the reader wondering how these findings are applicable and where researchers/managers might benefit and move forward.

R= We also thank and appreciate this reviewer’s observation, and we have considered and modified as suggested, our discussion.

6. The authors have generally written a nice manuscript, but it could be improved with some editing to correct grammar and improve sentence flow. A few examples are:

a. Lines 33 – 35: the first “difference” would read better as “differences”; “these different body sizes canid species” perhaps instead could be “these different sized canid species”

b. Line 47 and 59: what is “them” referencing?

c. Line 55: “its distribution the overlap of its diets” – what is “its” referencing?

d. Lines 61 – 65 and lines 73 - 77: these sentences are a bit hard to work through and could be improved with grammatical changes and/or being broken into two sentences.

R= we have done all suggested changes and also we reviewed our manuscript to edit and correct any grammar inconsistencies and tried to improve more our sentence flow.

Reviewer #2

The subject is interesting. However, it lacks many details of the methodology in the collection of samples. It is important to have information on the period with which the transects traveled, if in all the collections they traveled the same transects. This is due to the fact that generalist and opportunistic species such as the coyote and the gray fox can vary their diet in reduced time and space.

R= Similar request was done by reviewer 1. We have added all this detail to our corrected version of the manuscript, including information of sampling effort, sampling periods and if the same transects were travelled always.

On the other hand, the morphological characterization of the excrements would be convenient to strengthen it with some chemical or molecular technique; because its morphology is similar and the probability of error is high.

R= We have now provided more detail on the procedure we followed to minimize the potential misidentification of feces, and the conservative procedure implemented to assign with most confidence a feces sample to coyote or gray fox. The molecular or chemical methods suggested by the reviewer are indeed different procedures to assign to species a collected feces, however we already review all collected feces and we not have the possibility or the resources to implement those methods.

On the other hand, it is important to include complementary material with the characteristics of the hair of the identified species because several are similar.

R= We have now provided more detail on the procedure we followed to identify food items found on feces samples. Regarding its suggestion to include complementary material (e.g. microscopic slides), all the microscopic slides of guarding hairs we prepared with hairs obtained from the feces samples, are part of an ongoing undergraduate thesis work and therefore cannot be publicly shared yet. Those slides were compared to published microscopic images of guarding hairs of different mammal species that are included in several books, whose references were already cited in our original manuscript and still are in our corrected version of the ms.

We appreciate a lot the comments and observations done by reviewer 2, and we thank him to let us know who he is: Dr. Octavio Monroy-Vilchis, a researcher we followed and respect.

---

## [Decision Letter · Decision Letter 1]

8 Oct 2021

PONE-D-21-19581R1Trophic niche overlap between coyotes and gray foxes in a temperate forest in Durango, Mexico PLOS ONE

Dear Dr. Valenzuela-Galván,

Thank you for submitting your manuscript to PLOS ONE. After careful consideration, we feel that it has merit but does not fully meet PLOS ONE’s publication criteria as it currently stands. Therefore, we invite you to submit a revised version of the manuscript that addresses the points raised during the review process.

The authors adequately addressed all the reviewers' comments. However, I agree with Rev 1 in that readability of the manuscript must be improved, and English grammar and structure thoroughly checked and revised. Once these issues are addressed, I will be happy to accept the manuscript. Some minor changes are suggested below (please revise the entire manuscript for other typos or errors).

We look forward to receiving your revised manuscript.

Kind regards,

Nicoletta Righini, PhD

Academic Editor

PLOS ONE

Journal Requirements:

**Additional Editor Comments (if provided):**

**Some minor revisions:**

L. 16: Resource partitioning, and especially dietary partitioning, is …..

L. 18: Mexico – without accent

L. 18: Better put ‘in Mexico’ after ‘widely distributed’

L. 20: HAVE not been

L. 40- understanding the ways in which species partition these resources

L. 46- IT is relatively common…

L. 61- ..THEIR coexistence..

L. 68- THESE canids

l. 70 – HAVE not been thoroughly studied

L. 77- a mechanism partly explaining

L. 127- We CHOSE

L. 128-131: ‘spp.’ must not be italicized, only the name of the genus goes in italics, e.g. *Pinus* spp.

L. 132- formal collection of FECAL samples

L. 134- however THEY all produce..

L. 135- careful to IDENTIFY feces

L. 138: IN the vicinity of the collection site (delete ‘DE’)

L. 145: Please rephrase (season considered and considered…)

L. 145: we assigned feces TO any..

L. 146- feces collected IN a particular season WERE representative…

L: 147: IN that period of the year.

L. 148-49: In THE laboratory….and washed THEM with water

L. 160: trough THEIR guarding hairs

L. 167: the overall and SEASONAL representation

L. 170: the percentage of feces that CONTAINS…and although IT DOES not necessarily…

L. 185: Mexico without accent

L. 190 : and that they DID NOT HAVE any issues in LETTING us do…

L. 194: FECAL samples

L. 201: we EXPLORED

L. 337: detected in coyote’s feces likely represent carrion…

L. 340: COINCIDE with the feeding patternS

L. 373-374- IN the area…..differences we found IN trophic diversity

L. 378: However, the relevance of each….

L. 379: IN the coyote’s than IN the gray fox diet

L. 379: Delete ‘and’ at the beginning of the sentence

L. 395: FECAL samples

L. 398-399: shouldn’t this be ‘ defecate several hours LATER..’? (not ‘before’)

L. 399: IN a different habitat type

L. 399- However, differences among habitats in the success…

L. 401: in accordance WITH our findings

L. 410 – FOR a better understanding

L. 411: can help minimize

L. 415: IN this Natural Protected Area

Reviewers' comments:

Reviewer's Responses to Questions

**Comments to the Author**

1. If the authors have adequately addressed your comments raised in a previous round of review and you feel that this manuscript is now acceptable for publication, you may indicate that here to bypass the “Comments to the Author” section, enter your conflict of interest statement in the “Confidential to Editor” section, and submit your "Accept" recommendation.

Reviewer #1: (No Response)

2. Is the manuscript technically sound, and do the data support the conclusions?

Reviewer #1: Yes

3. Has the statistical analysis been performed appropriately and rigorously? 

Reviewer #1: Yes

4. Have the authors made all data underlying the findings in their manuscript fully available?

Reviewer #1: Yes

5. Is the manuscript presented in an intelligible fashion and written in standard English?

Reviewer #1: Yes

6. Review Comments to the Author

Reviewer #1: The authors thoughtfully addressed the initial review comments. My only minor comment is that the manuscript requires a final proof-read to eliminate any typos and improve readability through correcting some grammar and sentence structure. A few examples from the Introduction, but not a thorough list, are below:

• Line 46: what is the subject of “is relatively common” ?

• Line 54 – 57: This sentence is a bit difficult to understand

• Line 57: “this kind” should be changed to “these kind” to match the plural of “interactions”

o Also on line 62: “this questions” should be changed to “these questions”

• Line 62: “choose” should be “chose”

• Line 63: there should not be a comma after “species”

7. PLOS authors have the option to publish the peer review history of their article (what does this mean?). If published, this will include your full peer review and any attached files.

Reviewer #1: No

---

## [Author Response · Author response to Decision Letter 1]

4 Nov 2021

DETAILED DESCRIPTION ON HOW DO WE ATTEND ALL COMMENTS AND SUGGESTIONS DONE BY THE ASSOCIATE EDITOR AND BY TWO REVIEWERS

Journal Requirements:

Please review your reference list to ensure that it is complete and correct.

R= We have done so; all cited references are in the reference list, that is complete and correct.

If you have cited papers that have been retracted, please include the rationale for doing so in the manuscript text, or remove these references and replace them with relevant current references. Any changes to the reference list should be mentioned in the rebuttal letter that accompanies your revised manuscript. If you need to cite a retracted article, indicate the article’s retracted status in the References list and also include a citation and full reference for the retraction notice.

R= we are not citing any retracted paper. We have not changed our previous reference list. We have done, only minor changes for some references (e.g. added page numbers for a reference that did not have that information).

Additional Editor Comments (if provided):

Some minor revisions:

L. 16: Resource partitioning, and especially dietary partitioning, is …..

L. 18: Mexico – without accent

L. 18: Better put ‘in Mexico’ after ‘widely distributed’

L. 20: HAVE not been

L. 40- understanding the ways in which species partition these resources

L. 46- IT is relatively common…

L. 61- ..THEIR coexistence..

L. 68- THESE canids

l. 70 – HAVE not been thoroughly studied

L. 77- a mechanism partly explaining

L. 127- We CHOSE

L. 128-131: ‘spp.’ must not be italicized, only the name of the genus goes in italics, e.g. Pinus spp.

L. 132- formal collection of FECAL samples

L. 134- however THEY all produce..

L. 135- careful to IDENTIFY feces

L. 138: IN the vicinity of the collection site (delete ‘DE’)

L. 145: Please rephrase (season considered and considered…)

L. 145: we assigned feces TO any..

L. 146- feces collected IN a particular season WERE representative…

L: 147: IN that period of the year.

L. 148-49: In THE laboratory….and washed THEM with water

L. 160: trough THEIR guarding hairs

L. 167: the overall and SEASONAL representation

L. 170: the percentage of feces that CONTAINS…and although IT DOES not necessarily…

L. 185: Mexico without accent

L. 190 : and that they DID NOT HAVE any issues in LETTING us do…

L. 194: FECAL samples

L. 201: we EXPLORED

L. 337: detected in coyote’s feces likely represent carrion…

L. 340: COINCIDE with the feeding patternS

L. 373-374- IN the area…..differences we found IN trophic diversity

L. 378: However, the relevance of each….

L. 379: IN the coyote’s than IN the gray fox diet

L. 379: Delete ‘and’ at the beginning of the sentence

L. 395: FECAL samples

L. 398-399: shouldn’t this be ‘ defecate several hours LATER..’? (not ‘before’)

L. 399: IN a different habitat type

L. 399- However, differences among habitats in the success…

L. 401: in accordance WITH our findings

L. 410 – FOR a better understanding

L. 411: can help minimize

L. 415: IN this Natural Protected Area

R= ALL 42 requested changes were done in the new version of the manuscript

Review Comments to the Author

Reviewer #1: The authors thoughtfully addressed the initial review comments. My only minor comment is that the manuscript requires a final proof-read to eliminate any typos and improve readability through correcting some grammar and sentence structure. A few examples from the Introduction, but not a thorough list, are below:

• Line 46: what is the subject of “is relatively common” ?

• Line 54 – 57: This sentence is a bit difficult to understand

• Line 57: “this kind” should be changed to “these kind” to match the plural of “interactions”

o Also on line 62: “this questions” should be changed to “these questions”

• Line 62: “choose” should be “chose”

• Line 63: there should not be a comma after “species”

R= ALL 6 requested changes were done in the new version of the manuscript.

Asides, we have carefully reviewed the manuscript and corrected how we were using the square brackets for the references and some minor misspelling errors.

---

## [Editor Report · Decision Letter 2]

8 Nov 2021

Trophic niche overlap between coyotes and gray foxes in a temperate forest in Durango, Mexico

PONE-D-21-19581R2

Dear Dr. Valenzuela-Galván,

We’re pleased to inform you that your manuscript has been judged scientifically suitable for publication and will be formally accepted for publication once it meets all outstanding technical requirements.

Kind regards,

Nicoletta Righini, PhD

Academic Editor

PLOS ONE

---

## [Editor Report · Acceptance letter]

19 Nov 2021

PONE-D-21-19581R2 

Trophic niche overlap between coyotes and gray foxes in a temperate forest in Durango, Mexico 

Dear Dr. Valenzuela-Galván:

I'm pleased to inform you that your manuscript has been deemed suitable for publication in PLOS ONE. Congratulations! Your manuscript is now with our production department. 

Kind regards, 

on behalf of

Dr. Nicoletta Righini 

Academic Editor

PLOS ONE